# Aconitate Decarboxylase 1 Deficiency Exacerbates Mouse Colitis Induced by Dextran Sodium Sulfate

**DOI:** 10.3390/ijms23084392

**Published:** 2022-04-15

**Authors:** Ho Won Kim, A-Reum Yu, Ji Won Lee, Hoe Sun Yoon, Byung Soo Lee, Hwan-Woo Park, Sung Ki Lee, Young Ik Lee, Jake Whang, Jong-Seok Kim

**Affiliations:** 1Myunggok Medical Research Institute, College of Medicine, Konyang University, Daejeon 35365, Korea; kimong104@naver.com (H.W.K.); kyoaor22@hanmail.net (A.-R.Y.); smileday0103@naver.com (J.W.L.); cordelia_sun@naver.com (H.S.Y.); 2Department of Ophthalmology, Konyang University Hospital and College of Medicine, Daejeon 35365, Korea; kannylee@naver.com; 3Department of Cell Biology, Konyang University College of Medicine, Daejeon 35365, Korea; hwanwoopark@konyang.ac.kr; 4Department of Obstetrics and Gynecology, Konyang University Hospital, Daejeon 35365, Korea; sklee@kyuh.ac.kr; 5Lee’s Biotech Co., 415, C Dong, 17 Techno 4-ro, Yuseong-gu, Daejeon 34013, Korea; yilee@kribb.re.kr; 6Korea Mycobacterium Resource Center (KMRC), Department of Research and Development, The Korean Institute of Tuberculosis, Osong 28158, Korea; whangjake@gmail.com

**Keywords:** ulcerative colitis, Acod1, itaconate, 4-octyl itaconate, DSS

## Abstract

Ulcerative colitis is a complex inflammatory bowel disorder disease that can induce rectal and colonic dysfunction. Although the prevalence of IBD in Western countries is almost 0.5% of the general population, genetic causes are still not fully understood. In a recent discovery, itaconate was found to function as an immune-modulating metabolite in mammalian immune cells, wherein it is synthesized as an antimicrobial compound from the citric acid cycle intermediate cis-aconitic acid. However, the association between the Acod1 (Aconitate decarboxylase 1)-itaconate axis and ulcerative colitis has rarely been studied. To elucidate this, we established a DSS-induced colitis model with Acod1-deficient mice and then measured the mouse body weights, colon lengths, histological changes, and cytokines/chemokines in the colon. We first confirmed the upregulation of Acod1 RNA and protein expression levels in DSS-induced colitis. Then, we found that colitis symptoms, including weight loss, the disease activity index, and colon shortening, were worsened by the depletion of Acod1. In addition, the extent of intestinal epithelial barrier breakdown, the extent of immune cell infiltration, and the expression of proinflammatory cytokines and chemokines in Acod1-deficient mice were higher than those in wild-type mice. Finally, we confirmed that 4-octyl itaconate (4-OI) alleviated DSS-induced colitis in Acod1-deficient mice and decreased the expression of inflammatory cytokines and chemokines. To our knowledge, this study is the first to elucidate the role of the Acod1-itaconate axis in colitis. Our data clearly showed that Acod1 deletion resulted in severe DSS-induced colitis and substantial increases in inflammatory cytokine and chemokine levels. Our results suggest that Acod1 may normally play an important regulatory role in the pathogenesis of colitis, demonstrating the potential for novel therapies using 4-OI.

## 1. Introduction

Itaconate was first isolated from *Aspergillus terreus* and has been utilized in industrial processes for decades as a precursor to polymer synthesis [1,2]. Recent surprising findings have revealed that itaconate is produced in activated macrophages and that aconitate decarboxylase 1 (Acod1, also known as immune-responsive gene 1 [Irg1]) synthesizes itaconate from tricarboxylic acid cycle (TCA cycle) intermediate cis-aconitate [3,4]. Classically activated macrophages undergo metabolic reprogramming to facilitate macrophage effector functions, including cytokine release and ROS production [5]. In general, when macrophages are activated, glycolysis is upregulated, whereas pathways such as oxidative phosphorylation (OXPHOS) are downregulated [5]. However, in the initial stages of macrophage activation, the TCA cycle and OXPHOS are upregulated and result in the accumulation of immunometabolites such as succinate, fumarate, and itaconate, which exhibit a wide range of immunoregulatory functions [6]. Itaconate exerts antimicrobial activity by inhibiting isocitrate lyase, methylisocitrate lyase, and propionyl-CoA carboxylase, the key pathway enzymes, which are essential for bacterial growth [7]. In addition, itaconate has various immunomodulatory functions, such as inhibiting succinate dehydrogenase, which plays multiple roles in inflammation, inhibiting glycolysis and the NLRP3 inflammasome, and activating NRF2 and ATF3, transcriptional regulators with an anti-inflammatory action [8,9,10]. Since Acod1 was shown to be associated with itaconate in activated macrophages, itaconate has emerged as a mitochondrial metabolite with antimicrobial as well as immunomodulatory properties.

Inflammatory bowel disease (IBD) constitutes a group of chronic disorders, including both ulcerative colitis and Crohn’s disease, which are characterized by chronic inflammation of the gastrointestinal tract. IBD patients share several symptoms, including weight loss, rectal bleeding, diarrhea, fever, and severe abdominal pain [11]. The prevalence of IBD in Western countries is almost 0.5% of the general population, and the environmental and genetic causes of IBD as well as the development of therapeutic agents [12] are thus urgently needed. Abnormal gut microbiota, immune response dysregulation, and genetic mutations are known to be associated with the pathogenesis of IBD [13]. Th1 and Th17 immune responses are the main molecules involved, and many cytokines, including IL-1β, IL-6, IL-17, TNF-α, and TGF-β, and chemokines, are associated with IBD [13,14]. In addition, NOD2 and autophagy-related genes IRGM2, ATG16L1, IL23R, and PTPN2 have been reported to be associated with the pathogenesis of Crohn’s disease [15,16,17]. Although many researchers have attempted to identify novel pathogenic factors associated with IBD, including genetic, microbial, and immune response factors, its pathogenesis remains largely unclear.

The relationship between the Acod1-itaconate axis and IBD has rarely been studied. Recently, Qian et al. reported the efficacy of dimethyl itaconate (DMI), a cell-permeable itaconate, for inhibiting the generation of colitis-associated colorectal cancer [18]. They also confirmed the inhibitory effect of DMI on DSS-induced colitis, and this protective effect was attributed to the inhibited secretion of IL-1β and CCL2 by intestinal epithelial cells and the inhibited recruitment of macrophages and myeloid-derived suppressor cells [18]. In contrast, reports have indicated that itaconate may exert cancer-promoting effects through cellular pathways regulated by the obesity-related hormone leptin. They showed that itaconate downregulates the gene expression of peroxisome proliferator-activated receptor gamma (PPARγ), a colorectal cancer suppressor, in M2-like macrophages and upregulates anti-inflammatory cytokines, suggesting that it is associated with worse clinical outcomes in colorectal cancer patients [19]. Therefore, the role of Acod1-itaconate in colitis-related IBD and colorectal cancer is largely unknown. In addition, the abovementioned reports focused on itaconate and not the Acod1 gene.

This study aimed to investigate the role of Acod1 by utilizing an Acod1-deficient mouse model of DSS-induced colitis and, in particular, to elucidate the importance of itaconate. We used a mild DSS-induced colitis model and observed changes in body weight, the disease activity index (DAI), colon length, and histology and analyzed inflammatory cytokines and chemokines. In addition, 4-octyl itaconate (4-OI), a cell-permeable itaconate, was injected into Acod1-deficient mice to confirm its importance in our colitis model. Studies have indicated that Acod1 protein and mRNA levels are upregulated in DSS-induced colitis. In Acod1-deficient mice, colon shortening accompanied by rapid weight loss and increased levels of inflammatory cytokines and chemokines were observed even after low-dose DSS treatment. In addition, the rapid colitis and increased inflammatory cytokine and chemokine levels were suppressed by the injection of 4-OI into Acod1-deficient mice.

## 2. Results

### 2.1. Acod1 Is Upregulated in DSS-Induced Colitis, and Acod1 Depletion Results in Worsened Symptoms

We herein first measured the RNA and protein levels of Acod1 in the colon to assess its association with DSS-induced colitis. As a result, it was confirmed that mRNA and protein of ACOD1 were upregulated in the colon of DSS-induced colitis compared to the control group (Figure 1A,B). These results indicate that Acod1 is associated with DSS-induced colitis. Next, we used Acod1-deficient mice (*acod1*^−/−^) to more accurately elucidate the relationship between Acod1 and colitis (Appendix A). To do this, 2.2% DSS was administered to *acod1*^−/−^ mice and control wild-type (WT, C57BL/6) mice for 7 days (Figure 1C). DSS, at a concentration of 2.2%, was used to induce mild colitis in wild-type mice. Differences in weight loss between *acod1*^−/−^ mice and WT mice became significant on day 5 and became more severe by the end of the experiment. In WT mice, the differences in weight loss between the DSS treatment group and the regulator water treatment group were observed only on the 7th day. The weights of the WT and *acod1*^−/−^ mice provided regular water did not differ (Figure 1D). During the course of DSS treatment, *acod1*^−/−^ mice displayed more severe rectal bleeding and diarrhea (data not shown) and markedly increased DAI scores than the WT mice (Figure 1E). The DAI was calculated for each animal according to the body weight, stool consistency, and blood in stool scores (see Methods). In *acod1*^−/−^ mice, no acod1 expression was detected in the colon upon DSS treatment (Appendix A). 

Colon shortening, a macroscopic parameter of the colons sampled on Day 7, did not differ between water-treated *acod1*^−/−^ mice and WT mice. In contrast, colon contraction was more pronounced in *acod1*^−/−^ mice treated with 2.2% than in WT mice treated with 2.2% DSS (Figure 2A). We performed histochemical analyses of the colonic tissue sections to assess the extents of inflammation, inflammatory cell infiltration, and crypt architecture damage. When compared to the colon mucosal tissues of water-treated controls, DSS-treated mice exhibited extensive inflammation, inflammatory cell infiltration, and crypt architecture distortion. Remarkably, the tissue sections of *acod1*^−/−^ mice treated with the DSS solution showed more inflammatory aggravation than WT mice, as determined by the increased infiltration of inflammatory cells into the mucosa and the collapsibility of the epithelium and crypt structures (Figure 2B,C). The MPO enzyme is produced mainly by neutrophils, and its activity indicates the degree of neutrophil infiltration in a given tissue. Following 7 days of 2.2% DSS treatment, the MPO activity in WT and *acod1*^−/−^ mice was increased and significantly differed from that in the water-treated controls. Nevertheless, the MPO activity in WT and *acod1*^−/−^ mice treated with 2.2% DSS did not significantly differ (Figure 2D).

### 2.2. Acod1 Gene Deletion Increases Inflammatory Cytokine and Chemokine Expression in DSS-Induced Colitis

DSS is known to trigger inflammation in the gut via high levels of inflammatory mediators, such as iNOS, IL-1β, and IL-6 [20]. To investigate whether Acod1 is involved in the production of inflammatory cytokines and chemokines in DSS-induced colitis, qPCR analysis was performed on colonic tissues obtained on Day 7 after DSS administration. The expression levels of proinflammatory cytokines (IL-1β, IL-6, and IL-12p40); the chemokines CXCL-1 and CXCR-2, and the inflammatory mediator COX-2 were higher in *acod1*^−/−^ mice than in WT mice (Figure 3A–C). The expression levels of TNF-α were increased by DSS administration but did not differ between *acod1*^−/−^ mice and WT mice (Figure 3D). Overall, these results demonstrate that Acod1 gene deletion exacerbates colon inflammation in a mouse model of colitis.

### 2.3. 4-Octyl Itaconate Alleviates DSS-Induced Colitis in Acod1-Deficient Mice

In macrophages, Acod1 converts cis-aconitate into itaconate in the inflammatory state; inhibits succinate dehydrogenase, which is important for the maintenance of macrophage inflammation; and activates Nrf2 to suppress the production of reactive oxygen species [21]. Our data confirmed that Acod1 plays an essential role in the DSS-induced colitis model, but whether colitis is exacerbated due to the absence of the Acod1 gene or its product itaconate is unknown. Therefore, 4-OI, a cell-permeable itaconate, was herein used to treat *acod1*^−/−^ mice. In our mild DSS-induced colitis model, 4-OI was intraperitoneally injected daily for 7 days (Figure 4A). Notably, *acod1*^−/−^ mice treated with DSS did not show a significant difference in body weight until day 5 regardless of the presence or absence of 4-OI, but the body weights of mice in the 4-OI-treated group did not decrease after 5 days and were thereafter maintained. In addition, the DAI score increased more slowly in the 4-OI treatment group than in the DSS alone group, reaching a lower maximum (Figure 4B,C).

The shortening of the colon length induced by DSS was also partially recovered by 4-OI treatment, and the extent of the inflammation, inflammatory cell infiltration, and crypt architecture distortion was also partially recovered by 4-OI treatment (Figure 5A–C). The MPO activity in the 4-OI treatment group was equal to that in our previously described *acod1*^−/−^ mice but 4-OI did not prevent MPO activity. These results suggest that the severe condition of DSS-induced colitis caused by Acod1 deficiency is due to itaconate deficiency rather than the deficiency of other unknown functions of Acod1.

### 2.4. 4-Octyl Itaconate Diminishes the Expression of Inflammatory Cytokines and Chemokines in an Acod1-Deficient Mouse Model of DSS-Induced Colitis

We confirmed that 4-OI alleviated the symptoms of DSS-induced colitis in *acod1^−/−^* mice. Therefore, we tried to determine whether 4-OI also inhibits DSS-induced inflammation in *acod1*^−/−^ mice. The expression levels of IL-1β, IL-6, IL-12p40, CXCL-1, and COX-2 decreased in the 4-OI treatment group compared to the DSS treatment group (Figure 6A–C). On the other hand, TNF-α was not affected (Figure 6D). These results suggest that 4-OI relieved colitis in *acod1*^−/−^ mice by suppressing the expression of inflammatory cytokines and chemokines induced by DSS.

## 3. Discussion

After finding that Acod1 is upregulated in activated macrophages and catalyzes the synthesis of itaconate from cis-aconitate, its functions and product have continually been studied. Itaconate mainly functions to suppress the excessive activity of macrophages by inhibiting succinate dehydrogenase and increasing the induction of Nrf2, a transcriptional activator of antioxidant genes [8,10]. Therefore, several studies have confirmed the efficacy of Acod1 and its product itaconate in various diseases by assessing anti-inflammatory functions. For example, the expression of Acod1 is increased in subjects with ischemia–reperfusion liver injury, and the importance of the Acod1-itaconate axis as well as the therapeutic efficacy of 4-OI have been revealed using Acod1-deficient mice [22]. In the same context, Acod1 expression was herein shown to be upregulated in the colons of wild-type mice with DSS-induced colitis. The increase in Acod1 expression may have been due to an inflammatory reaction caused by DSS, but the specific cell types in which Acod1 expression was increased are not known. Perhaps the expression of Acod1 was increased in colonic macrophages or dendritic cells, but direct evidence of this is lacking, and future studies are needed to elucidate the cell types with increased Acod1 expression.

DSS colitis in mice most often begins with the loss of epithelial barrier function and the entry of luminal microorganisms or their products into the lamina propria, mostly due to the direct cytotoxicity of DSS. This entry results in the stimulation of innate and adaptive immune cells and the secretion of proinflammatory cytokines and chemokines. At the same time, it results in the influx of immune cells with cytotoxic potential, such as activated macrophages and neutrophils. Interestingly, the fact that DSS-induced colitis occurs even in SCID and *Rag1**^−/−^* mice lacking T cells mediating adaptation indicates that innate immunity plays a pivotal role in this colitis model [20]. In general, macrophages are known to be major secretion sources of proinflammatory cytokines and to play an important role in epithelial barrier function and proliferation, and neutrophils are known to contribute to tissue damage in the DSS colitis model. Herein, although the extents of immune cell infiltration and epithelial and crypt structural damage were higher in Acod1-deficient mice treated with 2.2% DSS than in wild-type mice, the MPO activity was slightly increased in the former group, but the difference was not statistically significant. This result was potentially due to the administration of a low concentration of DSS for 7 days. Because most neutrophils cause tissue damage in the early stage of inflammation, significant differences between wild-type and Acod1-deficient mice may not have been observable at the end of the experiment. This result may have been attributed to mild colitis being induced by the administration of low-dose DSS for 7 days, and Acod1 deficiency may not have had a significant effect on neutrophil infiltration. However, CXCL-1, a neutrophil chemotactic chemokine [23], exhibited a significantly increased mRNA expression level in Acod1-deficient mice treated with DSS, compared to wild-type mice, and the neutrophil counts should have been increased accordingly. Similarly, in Acod1-deficient mice treated with 4-OI after DSS induction, the MPO activity was not significantly changed, while CXCL-1 expression was decreased. Therefore, the correlation between neutrophils and CXCL-1 is not consistent in this model. However, the cause of this differential result will need to be elucidated in more detail in the future.

In the DSS-induced colitis model, inflammatory cytokines such as IL-1β, IL-6, and IL-12p40 were expressed at significantly higher levels in Acod1-deficient mice than in wild-type mice, whereas TNF-α levels did not differ between the two groups. The result was probably attributed to itaconate suppressing the secondary transcriptional response during LPS treatment in macrophages but not due to global inhibition of NFκB-dependent gene expression [8]. LPS transcriptionally regulates TNF-α and NFκbiz during the primary transcriptional response that activates NF-κB through the Toll-like receptor [24]. NFkbiz causes a secondary transcriptional response through iκbζ, and IL-6 and IL-12 are generated at this time [24,25]. Therefore, itaconate cannot inhibit the production of TNF-α among inflammatory cytokines produced by activated macrophages.

Wang et al. reported that DMI inhibits the development of colitis-associated colorectal cancer [18]. In this paper, DSS-induced colitis was confirmed to be suppressed in wild-type mice treated with DMI. However, according to our unpublished data, 4-OI had no inhibitory effect on colitis following the DSS treatment of wild-type mice, and 4-OI partially suppressed DSS-induced colitis in only Acod1-deficient mice (Figure 4, Figure 5 and Figure 6). Because itaconate has low cell permeability, a form of itaconate with high cell permeability has been developed. While DMI is the first cell-permeable itaconate to be developed, it does not exert its effects by directly penetrating cells but rather by increasing the biosynthesis of itaconate by some unknown mechanism and thus has no cell permeability [26]. In fact, 4-OI is an itaconate developed in a cell-permeable form and has therefore been used mainly in studies on the anti-inflammatory properties of itaconate. For example, 4-OI has shown efficacy in acute lung injury following LPS treatment and been shown to protect against systemic lupus via Nrf2 [27]. The differences in our experimental results and those reported by Qian are probably due to the different treatment agents (DMI and 4-OI) and to the concentrations of DSS treatment being slightly different.

According to our data, although 4-OI was treated, DSS-induced colitis in ACOD1-deficient mice was not completely inhibited. 4-OI only played a role in slowing the progression of colitis. These results are similar to the report that 4-OI can partially inhibit LPS-induced acute lung injury [28]. On the other hand, it is different from the result that the necrotic area caused by hepatic ischemia–reperfusion (I/R) in ACOD1-deficient mice was completely inhibited by 4-OI treatment [22]. The main mechanism of the mouse DSS-induced colitis model is primarily the loss of barrier function in intestinal epithelial cells and the resulting inflammatory reactions caused by invasion of luminal organisms or their by-products. In our model, 4-OI acted at a level that slows the progression of some DSS-induced colitis probably because 4-OI did not significantly affect the loss of the intestinal epitaxial barrier, even though it played a role in inhibiting inflammation. In addition, it cannot be ruled out that a significant increase in DSS-induced colitis in the absence of ACOD1 deficiency is not just due to the absence of itaconate but that another mechanism may also exist. A clearer related mechanism should be studied in the future.

Our data revealed that Acod1 deletion resulted in severe DSS-induced colitis and clearly demonstrated that the levels of inflammatory cytokines and chemokines were increased. In addition, the colitis symptoms of Acod1-deficient mice were alleviated by treatment with 4-OI. Our results suggest that Acod1 deficiency is due to a specific genetic defect in subjects with colitis, demonstrating the potential for novel therapies using 4-OI.

## 4. Materials and Methods

### 4.1. Animals

Acod1-deficient mice (C57BL/6NJ-Acod1^em1(IMPC)J^/J: *a**cod1*^−/−^) on a C57BL/6 background were purchased from Jackson Laboratories (Bar Harbor, ME, USA) and bred at our animal facility. Then, the offspring mice were examined by genotyping PCR, which was achieved by standard PCR amplification of genomic DNA isolated from mouse ears using a Tissue Direct PCR Kit (Omega BioTek, GA, USA). PCR primers provided by Jackson Laboratories were used for mouse genotyping. The following primers were used: common (5′-GTGGGGAGGGGAACTATGAG-3′), wild-type reverse (5′-ATTTGGAGGAACCCCATGAC-3′), and mutant reverse (5′-CAGCCTCTAAGCCAGACAGC-3′, Appendix A). C57BL/6 mice were purchased from DBL (Chungcheongbuk-do, Korea). All mice used were 8 weeks old. The animals were housed in a specific pathogen-free (SPF) room under controlled conditions of a 12 h light-dark cycle and a constant temperature (25 °C). Animal Care and the Guiding Principles for Animal Experiment Using Animals were approved by the University of Konyang Animal Care and Use Committee (20-01-E-01).

### 4.2. DSS induces Colitis in Mice

C57BL/6 mice (wild-type) and *acod1*^−/−^ mice (male, 8 weeks) were used for the mouse colitis model. The murine DSS-induced colitis model was performed as previously described, with slight modifications [29]. Briefly, DSS-treated mice were prepared by the administration of 2.2% DSS (molecular weight 36–50 kDa, MP Biomedicals, Santa Ana, CA, USA) to their drinking water, to which they had ad libitum access to for 7 days. Control animals were supplied regular drinking water. The mice were monitored every day for 7 days to measure the DAI. Surviving mice were sacrificed after 7 days, and colonic tissues were collected. In some experiments, mice were intraperitoneally injected with 4-OI (Medchemexpress, NJ, USA).

### 4.3. Quantification of mRNA by Real-Time RT–PCR

Total RNA was isolated from the colon tissues of mice using an AccuPrep™ Universal RNA Extraction Kit (BIONEER, Taejon, Korea), in accordance with the manufacturer’s protocol. Then, 500 ng of total RNA was reverse-transcribed into single-strand cDNA with PrimeScript RT Master Mix (TaKaRa, Japan). Real-time PCR was performed using LightCycler^®^ 480 SYBR^®^ Green Ⅰ Master Mix (Roche, catalog no. 04707516001) on a CFX Connect™ Real-Time system, and the results were analyzed with CFX Maestro™ software 1.1. The real-time PCR protocol began with an initial enzyme activation step (10 min, 94 °C), followed by 40 cycles consisting of denaturing (30 sec, 94 °C) and annealing/extending (30 sec, 60 °C). The primer sets used were as follows: mouse IL-1β (forward: GCCCATCCTCTGTGACTCAT; reverse: AGGCCACAGGTATTTTGTCG), IL-6 (forward: GATGGATGCTACCAAACTGGAT; reverse: CCAGGTAGCTATGGTACTCCAGA), IL-12p40 (forward: GGAAGCACGGCAGCAGAATA; reverse: AACTTGAGGGAGAAGTAGGAATGG), COX-2 (forward: AACCGCATTGCCTCTGAAT; reverse: CATGTTCCAGGAGGATGGAG), TNF-α (forward: TCTTCTCATTCCTGCTTGTGG; reverse: GGTCTGGGCCATAGAACTGA), CXCL-1 (forward: ATCCAGAGCTTGAAGGTGTTG; reverse: GTCTGTCTTCTTTCTCCGTTACTT) and CXCR-2 (forward: CAGCGACCCAGTCAGGATTTA reverse: ACCAGCATCACGAGGGAGTTT). The amount of RNA was normalized to the β-actin signal amplified in a separate reaction (forward primer: TACCCAGGCATTGCTGACAGG; reverse: ACTTGCGGTGCACGATGGA).

### 4.4. Disease Activity Index

The DAI was calculated for each animal by adding the scores of body weight loss, stool consistency, and blood in stool, as described below. Body weight loss scores: 0, no loss; 1, 1~5% loss of body weight; 2, 5~10% loss of body weight; 3, 10~20% loss of body weight; and 4, more than 20% loss of body weight. Stool consistency scores: 0, normal feces; 1, loose stool; 2, watery diarrhea; 3, slimy diarrhea and little blood; and 4, severe watery diarrhea with blood. Blood in stool scores: 0, no blood; 2, blood in stool; and 4, outflow of blood from the anus.

### 4.5. Histological Analysis and Scoring of Colonic Damage

Tissues were fixed in 10% PBS-buffered formalin overnight and then embedded in paraffin wax. For injury assessment, 6-μm-thick sections were stained with hematoxylin and eosin (H&E). Histological images taken using a 4× or 40× objective on H&E-stained tissue sections of colon (Leica dm500, Heerbrugg, Switzerland).

Histological scoring was based on three independent parameters: extent of inflammation (0~3: none, mucosal, submucosal, and transmural); extent of inflammatory cell infiltration (0~3: none, mild, moderate, and severe); and crypt architecture damage (0~2: none, regeneration, and destruction). Combining all scores for the individual parameters resulted in a total score ranging from 0 to 8.

### 4.6. Determination of Myeloperoxidase (MPO) Activity

The MPO activity in colonic tissues was measured using an MPO assay kit (BioVision, catalog no. K744-100). The colon tissue was homogenized in four volumes of MPO Assay Buffer and then centrifuged at 13,000× *g* for 10 min to remove insoluble material. Fifty microliters of each sample were used for the assay after pellet lysis using 200 μL of MPO Assay Buffer. The MPO activity in the samples was calculated from the standard curve.

### 4.7. Statistical Analysis

All comparisons of three or more treatment groups were determined by one-way ANOVA, followed by Tukey’s multiple comparisons test. The data are presented as the means ± SDs. GraphPad Prism^®^ version 6.01 (GraphPad Software, San Diego, CA, USA) was used for all calculations. *p* values less than 0.05 indicated statistical significance.

## 5. Conclusions

In conclusion, these results suggest that that Acod1 deletion resulted in severe DSS-induced colitis and substantial increases in inflammatory cytokine and chemokine levels. In addition, the colitis symptoms of Acod1-deficient mice were alleviated by treatment with 4-OI. Our results suggest that Acod1 may normally play an important regulatory role in the pathogenesis of colitis, demonstrating the potential for novel therapies using 4-OI.

## Figures and Tables

**Figure 1 ijms-23-04392-f001:**
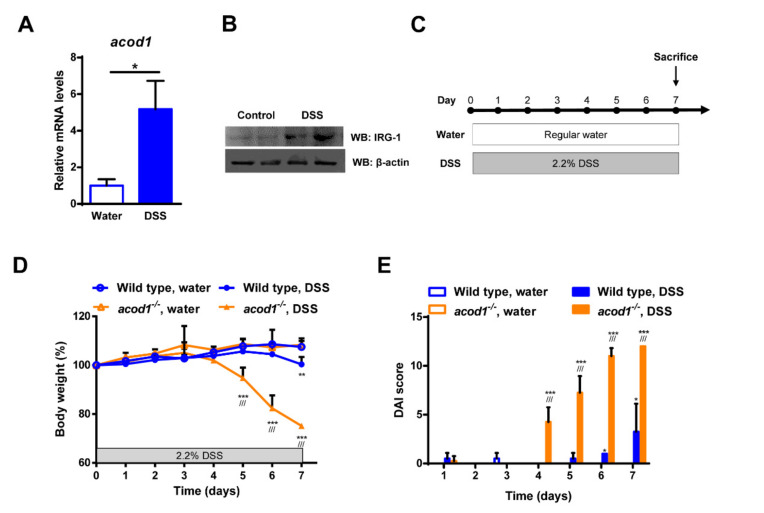
The effects of Acod1 gene deficiency on the clinical features of DSS-induced colitis in C57BL/6 mice. The colonic mRNA (**A**) and protein expression (**B**) levels of Acod1 in WT mice treated with DSS. (**C**) A schematic illustration of the mouse model of DSS-induced colitis. WT and *acod1*^−/−^ mice were administered regular water or 2.2% DSS for 7 days. (**D**) Weight changes in WT and *acod1*^−/−^ mice within 7 days of treatment with DSS. (**E**) Disease activity index (DAI) scores of WT and *acod1*^−/−^ mice after DSS treatment. *(** *p* < 0.05, ** *p* < 0.01, and *** *p* < 0.001 compared with WT. *///p* < 0.001 WT + DSS versus *acod1*^−/−^ + DSS). (**D**) Colonic mRNA levels of Acod1 in WT mice treated with DSS.

**Figure 2 ijms-23-04392-f002:**
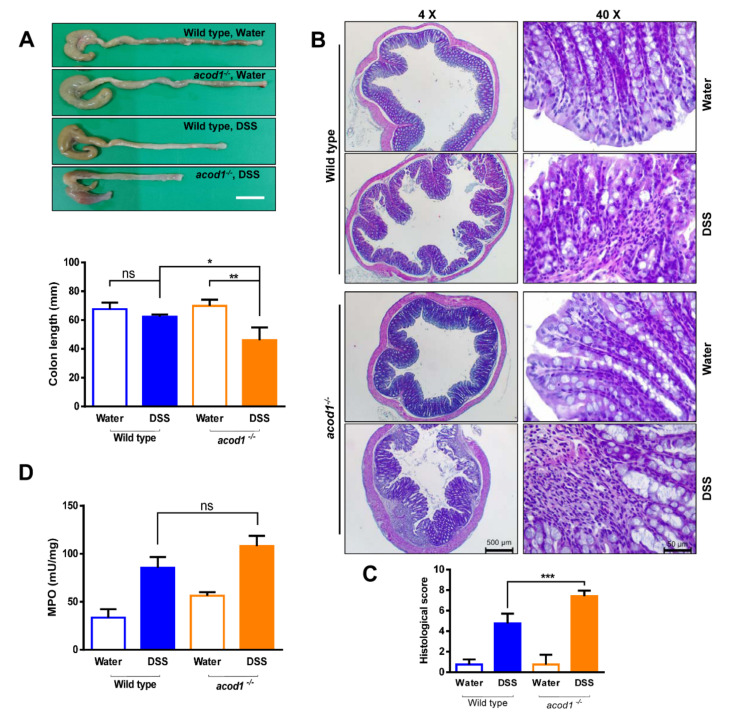
Acod1 gene deficiency aggravated colon contraction induced by DSS. (**A**) Representative images of colons isolated from WT and *acod1*^−/−^ mice treated with regular water or 2.2% DSS for 7 days. The colon lengths in each group were measured at 7 days (n = 6–8). * *p* < 0.05 WT+DSS versus *acod1*^−/−^ + DSS. ** *p* < 0.01 *acod1*^−/−^ versus *acod1*^−/−^ + DSS. (**B**) Representative histopathological images of colon tissue sections from water- and DSS-treated WT and *acod1*^−/−^ mice. Left panel images taken using 4× objective on an H&E-stained tissue section (Scale bar: 500 um). Right panel images taken using 40× objective on an H&E-stained tissue section (Scale bar: 50 um). (**C**) Histological scores of the different groups of mice. *** *p* < 0.001 WT+DSS versus *acod1*^−/−^ + DSS. (**D**) Neutrophil infiltration into the colon on day 7 as quantified by measuring the myeloperoxidase activity in accordance with the manufacturer’s protocol (ns, not significant).

**Figure 3 ijms-23-04392-f003:**
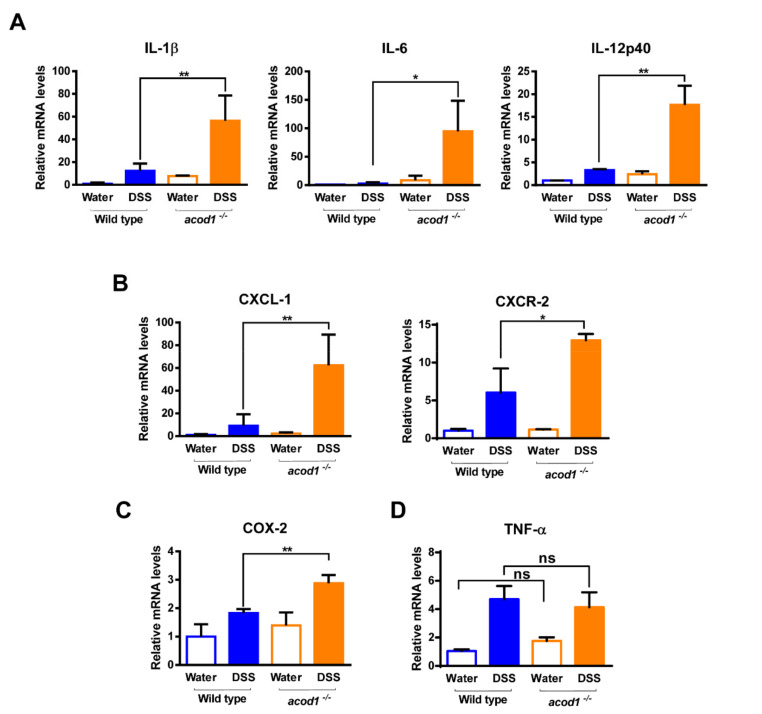
Acod1 gene deficiency induces inflammatory cytokine, the inflammatory mediator, and chemokine expression in DSS-induced colitis. The mRNA levels of IL-1β, IL-6, and IL-12p40 (**A**); CXCL-1, CXCR-2 (**B**), COX-2 (**C**); and TNF-α (**D**) in the colonic tissues were determined by quantitative real time-PCR. Expression was normalized to that of β-actin, and each bar represents the mean ± SD. Symbols * and ** denote *p* < 0.05 and *p* < 0.01, respectively.

**Figure 4 ijms-23-04392-f004:**
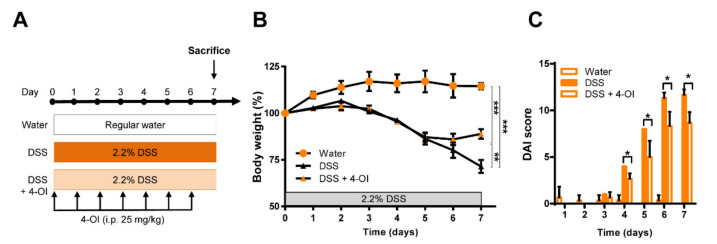
The effects of 4-octyl itaconate (4-OI) on DSS-induced colitis in *acod1*^−/−^ mice. (**A**) The experimental scheme for the administration of DSS and 4-OI. The 4-OI agent was injected intraperitoneally daily at 25 mg/kg for 7 days. (**B**) The administration of 4-OI reduced the body weight loss caused by DSS-induced colitis in *acod1*^−/−^ mice. Symbols ** and *** denote *p* < 0.01 and *p* < 0.001, respectively. (**C**) The administration of 4-OI reduced the DAIs of mice with DSS-induced colitis. ** p* < 0.05 *acod1*^−/−^ mice (n = 6) + DSS versus *acod1*^−/−^ mice + DSS + 4-OI (n = 6).

**Figure 5 ijms-23-04392-f005:**
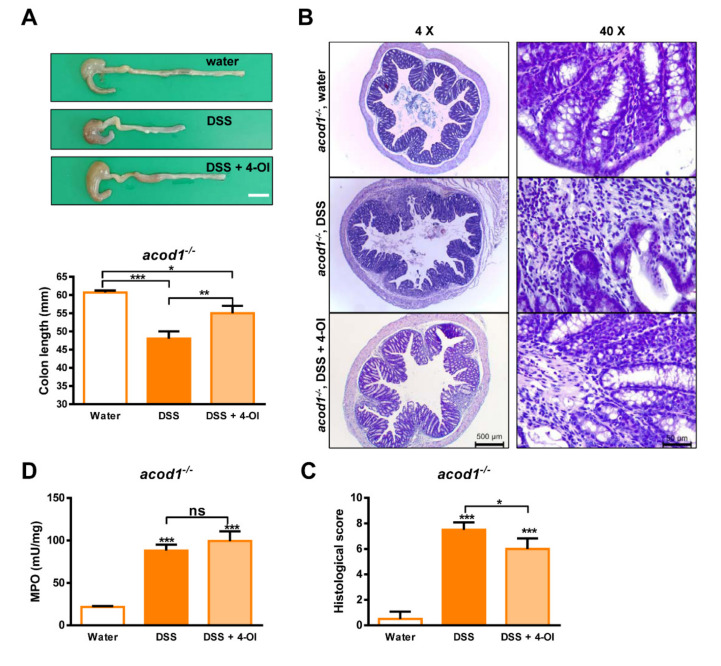
The effects of 4-octyl itaconate (4-OI) on the clinical features of *acod1^−/−^* mice with DSS-induced colitis. (**A**) Morphological changes in the colons of *acod1^−/−^* mice with DSS-induced colitis treated with or without 4-OI. The colon lengths in each group were measured on Day 7 (n = 6–8). Representative histological images of colon sections (**B**) and histological scores (**C**) of *acod1*^−/−^ mice with DSS-induced colitis treated with or without 4-OI. Left panel images taken using 4× objective on an H&E-stained tissue section (Scale bar: 500 um). Right panel images taken using 40× objective on an H&E-stained tissue section (Scale bar: 50 um). (**D**) Myeloperoxidase activity was measured to quantify neutrophil infiltration in the colon. Symbols *, **, and *** denote *p* < 0.05, *p* < 0.01, and *p* < 0.001, respectively.

**Figure 6 ijms-23-04392-f006:**
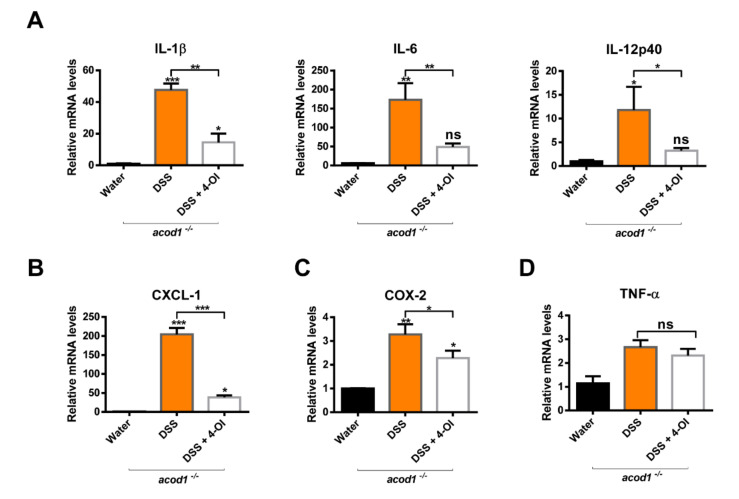
Effects of 4-octyl itaconate (4-OI) on inflammatory cytokine expression in *acod1*^−/−^ mice with DSS-induced colitis. The mRNA levels of IL-1β, IL-6, and IL-12p40 (**A**); CXCL-1 (**B**); COX-2 (**C**); and TNF-α (**D**) in the colonic tissues were determined by quantitative real time-PCR. Expression was normalized to that of β-actin, and each bar represents the mean ± SD. Symbols *, **, and *** denote *p* < 0.05, *p* < 0.01, and *p* < 0.001, respectively.

## Data Availability

Any data or material that support the findings of this study can be made available by the corresponding author upon request.

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
