# Peer review of "Aconitate Decarboxylase 1 Deficiency Exacerbates Mouse Colitis Induced by Dextran Sodium Sulfate"

_ijms, 2022, doi:10.3390/ijms23084392_

Round 1
Reviewer 1 Report
Please see the attachment.

Author Response
April 1, 2022
Manuscript ID: ijms-1626097
Manuscript Title: Aconitate decarboxylase 1 deficiency exacerbates mouse colitis induced by dextran sodium sulfate
Dear Reviewer
We are tremendously grateful for your generosity. We thank you for granting us time to revise our manuscript. As a result of your thoughtful consideration, we were able to conduct additional experiments and prepare new data. We also appreciate that you and the reviewers have spent valuable time to provide us with expert comments on our manuscript. We want to submit our revised manuscript, in which we believe we have addressed each concern raised by the reviewers. As suggested, we have reviewed the text and figures, and we have attempted to strike a proper balance between brevity and completeness.
We hope that the clarity of the revised manuscript has improved and that this alleviates any concerns you may have regarding the quality of our data. The reviewers' comments are addressed individually below, and we have indicated the specific alterations (page and line) that were made in each response.
We look forward to any additional comments or questions you may have on this manuscript, and we hope that the revised version is now acceptable for publication in IJMS.
Yours truly,
Jong-Seok Kim, on behalf of all of the authors.
Assistant Professor
Myunggok Medical Research Institute, College of Medicine, Konyang University, 158 Seogu Gwanjudongro Daejeon, 35365, South Korea.
E-mail: jskim7488@konyang.ac.kr
Tel.: +82-42-600-8648
Reviewer’s comments:
In this present article, the authors presented the results of a very interesting study of aconitate decarboxylase (ACOD1)-deficient in relation to mouse colitis induced by dextran sodium sulfate. Although, the study design is strong and impressive, but the data and the results presentation are poor. Overall, the study is nice and timely, and well written. However, there are several major concerns need to be addressed.
[A] We appreciate the reviewer’s clear summary of our work and these valuable comments.
Major comments:
- 1, ACOD1 KO mice need more characterization, showing that ACOD 1 is deleted in the colon through IHC, RT-qPCR, Western blot (WB) etc., without which the results are incomplete. Authors didn’t show WB of Acod1-/- along with control. Fig. 1B, must requires addition of Acod1-/-.
[A] We fully agree with the reviewer's concerns. Nevertheless, the reason for showing Figure 1A and 1B in the purpose and flow of our study is to confirm the relevance of ACOD1 in colitis caused by DSS in wild-type mice. In order to find the association of interesting gene in a particular disease model, as you know, it starts with measuring the expression level of genes of interest. Therefore, we first measured the expression level of ACOD1 upon induction of colitis with a pilot test and then confirmed that it increased. After confirming the increase, we purchased ACOD1 KO mice from Jackson Labs. While we agree with the reviewers' concerns, we have not been able to buy and use mice with specific gene deletions before establishing an association with disease. Therefore, when making the protein sample in Figure 1B, there was no ACOD1 KO mouse, and unfortunately, in order to produce the requested results, it is necessary to conduct an animal experiment again. However, please consider that the deadline for submitting the revised manuscript is very short. Fortunately, we will add RT-qPCR data that may alleviate some of the reviewer's concerns. The cDNA is a sample to measure cytokines after purchasing ACOD1 KO mice. Please, kindly consider our situation. The RT-qPCR data were included as a supplemental figure in the flow of the manuscript.
- 2, colitis being an inflammatory disease, it is very hard to see any inflammatory signs other than the length of the colons which are pixel wise not good. Likewise, IHC results of colonic sections are of very poor quality. Instead of regional blowup, the author should use at least 20X lens to capture images. These images do not tell anything about immune infiltration, colon integration etc. It requires total revise.
[A] We fully agree with the reviewer's comment. We used an inverted microscope to generate the data. We quickly bought a tissue microscope (Leica) and replaced it with better quality data. We have added changed data to the revised manuscript. We appreciated your valuable comment.
- 5, Overall, the image quality must improve especially in Fig. 5B. The image labelling seems not authentic. What do authors mean 40X & 100X? It looks upper panel images were taken by 4X lens, not 40X and lower panel image were taken by 10X lens. Most of all, the image quality is not acceptable.
[A] We fully agree with the reviewer's comment. We have added changed data to the revised manuscript.
- All the bar graphs’ line thickness should increase, especially the error bars.
[A] Corrected.
Minor comments:
It requires the details of image acquisition process.
[A] We have added the methodologic details to the revised manuscript (Page 5, Line 152-154; Page 8, Line 209-211; Page 11, Line 379-380).

Reviewer 2 Report
Kim et al. compared symptoms of DSS-induced colitis in mice of either wild type (WT) or aconitate decarboxylase (Acod) 1-deficient (acod1-/-) genotype. Engineered lack of Acod1 expression, which converts aconitate into itaconate and is enhanced in WT colons after DSS treatment, aggravated DSS-induced clinical symptoms and parameters of colitis. Colonic expression of inflammatory mediators is induced by DSS treatment and enhanced due to Acod1 deficiency, with the exception of TNF, which is refractory to Acod1 deficiency. The application of 4-octyl itaconate (4-OI), a cell-permeable derivate of itaconate partly reversed the DSS-induced phenotype in acod1-/- mice. The authors conclude that Acod1 has an important role colitis pathogenicity and that 4-OI may serve as a molecular core for the development of new therapeutic options.
While the data provided are of potential interest, some details have to be addressed prior to acceptance for publication.
- Line 25: defince Acod1.
- Line 27: rephrase.
- Lines 67 – 69 (…, and … [12]. …): rephrase.
- Line 70: I believe the etiology is generally accepted, but not the uncontrolled immune response: rephrase.
- Lines 107 – 122: This paragraph is somewhat confusing, please re-organize.
- Line 160: pPCR probably should read qPCR.
- Line 168: Acod1 gene deficiency does not induce inflammatory cytokine etc. expression, but rather increases it. Probably, only TNF is induced by Acod1 deficiency, but this has to be proved by statistical analysis.
- Lines 172 – 173 + 222 - 223: This is description of a result, in contrast to the other information of the figure legends, and has not to appear at this place.
- Lines 187 – 189: No, looking at the figure, it appears that DAI does not gradually improve, but that it increases slower and reaches a lower maximum.
- Most importantly, the amelioration observed after 4-OI application is much less prominent than the aggravation due to Acod1 deficiency. Thus, one cannot exclude other mechanisms than itaconate absence at the basis of the acod1-/- This has to be emphasized and adequately discussed.
- Lines 214 – 215: the relative clause ‘, such as … and CXCl1,’ should appear directly after ‘chemokines’, but not after ‘COX-2’.
- Lines 231/232: what is an ‘antioxidant transcription gene’?
- Lines 268 – 270: The potential acod-/--induced alteration of CXCR2 most probably affects its expression, which easily could/should be analyzed to support this hypothesis.
- Lines 270 – 279: The difference in responsiveness between TNF and the other meediators is quite interesting. May this observation provide some indication on the target of itaconate? Please, discuss.
- Lines 79 + 280 + 293: The citation ‘Qian et al.’ probably should read ‘Wang et al.’.
- Lines 299 – 300: I do not understand the meaning of the first clause, thus the reasoning for the conclusion in the second clause. Please, rephrase.
Author Response
April 1, 2022
Manuscript ID: ijms-1626097
Manuscript Title: Aconitate decarboxylase 1 deficiency exacerbates mouse colitis induced by dextran sodium sulfate
Dear Reviewer
We are tremendously grateful for your generosity. We thank you for granting us time to revise our manuscript. As a result of your thoughtful consideration, we were able to conduct additional experiments and prepare new data. We also appreciate that you and the reviewers have spent valuable time to provide us with expert comments on our manuscript. We want to submit our revised manuscript, in which we believe we have addressed each concern raised by the reviewers. As suggested, we have reviewed the text and figures, and we have attempted to strike a proper balance between brevity and completeness.
We hope that the clarity of the revised manuscript has improved and that this alleviates any concerns you may have regarding the quality of our data. The reviewers' comments are addressed individually below, and we have indicated the specific alterations (page and line) that were made in each response.
We look forward to any additional comments or questions you may have on this manuscript, and we hope that the revised version is now acceptable for publication in IJMS.
Yours truly,
Jong-Seok Kim, on behalf of all of the authors.
Assistant Professor
Myunggok Medical Research Institute, College of Medicine, Konyang University, 158 Seogu Gwanjudongro Daejeon, 35365, South Korea.
E-mail: jskim7488@konyang.ac.kr
Tel.: +82-42-600-8648
Kim et al. compared symptoms of DSS-induced colitis in mice of either wild type (WT) or aconitate decarboxylase (Acod) 1-deficient (acod1-/-) genotype. Engineered lack of Acod1 expression, which converts aconitate into itaconate and is enhanced in WT colons after DSS treatment, aggravated DSS-induced clinical symptoms and parameters of colitis. Colonic expression of inflammatory mediators is induced by DSS treatment and enhanced due to Acod1 deficiency, with the exception of TNF, which is refractory to Acod1 deficiency. The application of 4-octyl itaconate (4-OI), a cell-permeable derivate of itaconate partly reversed the DSS-induced phenotype in acod1-/- mice. The authors conclude that Acod1 has an important role colitis pathogenicity and that 4-OI may serve as a molecular core for the development of new therapeutic options.
While the data provided are of potential interest, some details have to be addressed prior to acceptance for publication.
[A] We appreciate the reviewer’s clear summary of our work and these valuable comments.
- Line 25: defince Acod1.
[A] Corrected [Page 1, Line 25].
- Line 27: rephrase.
[A] Corrected [Page 1, Line 26-28].
- Lines 67 – 69 (…, and … [12]. …): rephrase.
[A] Corrected [Page 2, Line 66-69].
- Line 70: I believe the etiology is generally accepted, but not the uncontrolled immune response: rephrase.
[A] Corrected [Page 2, Line 69-70].
- Lines 107 – 122: This paragraph is somewhat confusing, please re-organize.
[A] Corrected [Page 3, Line 107-123].
- Line 160: pPCR probably should read qPCR.
[A] Corrected [Page 5, Line 163].
- Line 168: Acod1 gene deficiency does not induce inflammatory cytokine etc. expression, but rather increases it. Probably, only TNF is induced by Acod1 deficiency, but this has to be proved by statistical analysis.
[A] We appreciate this comment by the reviewer. However, based on our experience in conducting several other experiments with ACOD1 KO, we speculated that the data must have been an experimental error. Therefore, to confirm this, RT-PCR experiments were performed again. As a result, it was confirmed that TNF-α did not increase with only ACOD1 deficiency. The result has been added to the revised manuscript. Sorry for the confusion (Figure 3D and Figure 6D).
- Lines 172 – 173 + 222 - 223: This is description of a result, in contrast to the other information of the figure legends, and has not to appear at this place.
[A] Corrected [Page 6, Line 174; Page 8, Line 224].
- Lines 187 – 189: No, looking at the figure, it appears that DAI does not gradually improve, but that it increases slower and reaches a lower maximum.
[A] We appreciate this comment by the reviewer. We fully agree with your comment. We corrected the sentence and wrote it in the revised manuscript (Page 6, Line 188-190).
- Most importantly, the amelioration observed after 4-OI application is much less prominent than the aggravation due to Acod1 deficiency. Thus, one cannot exclude other mechanisms than itaconate absence at the basis of the acod1-/-This has to be emphasized and adequately discussed.
[A] We appreciate your valuable comment. The discussion is described in the revised manuscript. (Page 10, Line 297-310).
Lines 214 – 215: the relative clause ‘, such as … and CXCl1,’ should appear directly after ‘chemokines’, but not after ‘COX-2’.
[A] Corrected (Page 8, Line 217-218).
- Lines 231/232: what is an ‘antioxidant transcription gene’?
[A] Corrected (Page 8, Line 233-234).
- Lines 268 – 270: The potential acod-/--induced alteration of CXCR2 most probably affects its expression, which easily could/should be analyzed to support this hypothesis.
[A] We appreciate this comment by the reviewer. We additionally confirmed the difference in colonic expression of CXCR-2 upon DSS induction in WT and ACOD1 deficiency mouse. The CXCR2 qPCR data were added to Figure 3 and the hypothesis removed.
- Lines 270 – 279: The difference in responsiveness between TNF and the other meediators is quite interesting. May this observation provide some indication on the target of itaconate? Please, discuss.
[A] We appreciate this comment by the reviewer. Honestly, it's hard to understand exactly what the reviewer question’s meaning. The earliest known target of itaconate is the inhibition of succinate dehydrogenase (Cell metabolism, 2016, volume 23, issue 1, Page 158-166). Similar to our results in this paper, TNF-α was not increased in acod1-deficient macrophages even though most of the inflammatory cytokines were increased during LPS treatment than in WT. In addition, it was confirmed that only the level of TNF-α was not changed upon treatment with dimethyl itaconate, indicating that itaconate was not due to global inhibition of NF-kB-dependent gene expression. This difference may be due to the fact that itaconate inhibits the NF-kB second signal, NFkbiz, rather than inhibiting the overall NF-kB signal, as already discussed in the discussion section. There may be another itaconate target, but our results can be explained only by the mechanisms identified in the previous paper.
Lines 79 + 280 + 293: The citation ‘Qian et al.’ probably should read ‘Wang et al.’.
[A] Corrected [Page 9, Line 282].
- Lines 299 – 300: I do not understand the meaning of the first clause, thus the reasoning for the conclusion in the second clause. Please, rephrase.
[A] We appreciate this comment by the reviewer. We've removed unnecessary and confusing sentences.

Round 2
Reviewer 1 Report
All the bar graphs' line thickness still needs improvement (Fig. 3, Fig. 4C, Fig. 5A, D &C, & Fig. 6).
Author Response
Reviewer’s comments:
All the bar graphs' line thickness still needs improvement (Fig. 3, Fig. 4C, Fig. 5A, D &C, & Fig. 6).
[A] At the reviewer's suggestion, the line thickness of all figures was changed from “1” to “2”.
